# *In Vitro* Evaluation of Ag- and Sr-Doped Hydroxyapatite Coatings for Medical Applications

**DOI:** 10.3390/ma16155428

**Published:** 2023-08-02

**Authors:** Elena Ungureanu, Alina Vladescu (Dragomir), Anca C. Parau, Valentina Mitran, Anisoara Cimpean, Mihai Tarcolea, Diana M. Vranceanu, Cosmin M. Cotrut

**Affiliations:** 1Faculty of Materials Science and Engineering, University Politehnica of Bucharest, 313 Independentei Street, 060042 Bucharest, Romania; elena.ungureanu1102@upb.ro (E.U.); mihai.tarcolea@upb.ro (M.T.); 2Department for Advanced Surface Processing and Analysis by Vacuum Technologies, National Institute of Research and Development for Optoelectronics—INOE 2000, 409 Atomistilor Street, 77125 Magurele, Romaniaanca.parau@inoe.ro (A.C.P.); 3Department of Biochemistry and Molecular Biology, University of Bucharest, 91-95 Independentei Street, 050095 Bucharest, Romania; valentina.mitran@bio.unibuc.ro (V.M.); anisoara.cimpean@bio.unibuc.ro (A.C.)

**Keywords:** hydroxyapatite, silver, strontium, electrochemical deposition, bioactivity, osteogenic differentiation, biocompatibility

## Abstract

Osseointegration plays the most important role in the success of an implant. One of the applications of hydroxyapatite (HAp) is as a coating for metallic implants due to its bioactive nature, which improves osteoconduction. The purpose of this research was to assess the in vitro behavior of HAp undoped and doped with Ag and/or Sr obtained by galvanostatic pulsed electrochemical deposition. The coatings were investigated in terms of chemical bonds, contact angle and surface free energy, electrochemical behavior, in vitro biomineralization in acellular media (SBF and PBS), and biocompatibility with preosteoblasts cells (MC3T3-E1 cell line). The obtained results highlighted the beneficial impact of Ag and/or Sr on the HAp. The FTIR spectra confirmed the presence of hydroxyapatite within all coatings, while in terms of wettability, the contact angle and surface free energy investigations showed that all surfaces were hydrophilic. The in vitro behavior of MC3T3-E1 indicated that the presence of Sr in the HAp coatings as a unique doping agent or in combination with Ag elicited improved cytocompatibility in terms of cell proliferation and osteogenic differentiation. Therefore, the composite HAp-based coatings showed promising potential for bone regeneration applications.

## 1. Introduction

The osseointegration capacity of a biomaterial is the main factor on which the success of an implant depends. This process can last up to 3 months, being divided into three major stages [1]. In the first 4–6 weeks after the surgical intervention, the formation of a primitive bone tissue that allows the connection between the bone walls and the implant takes place. This stage occurs due to mesenchymal cell proliferation, osteoinduction, and osteoconduction. Later, the primitive bone tissue begins to change into a lamellar bone tissue that confers mechanical resistance. The last stage of the osseointegration process is correlated with bone remodeling and involves osteoclast-mediated bone resorption. This stage starts from the third month after implantation and continues throughout life [2,3]. However, this process depends on factors such as age [4,5], gender [6], race [7], and comorbidities [8].

Osteoporosis is one of the diseases faced by most women over 50 years old [9]. According to a 2019 report on osteoporosis in the European Union, the number of women suffering from osteoporosis is four times higher than the number of men. At the European level, osteoporosis affects approximately 22% of the population of each country [10]. Regarding bone tissue, it can be said that osteoporosis is a disease that affects the lifestyles of elderly patients, and it is one of the reasons why the osseointegration process is slowed down.

Another problem that can affect the osseointegration capacity of implants is the occurrence of infections. In recent years, the high resistance of bacteria to antibiotics has become a threat to the lives of patients who have undergone surgery [11]. Infections associated with orthopedic and dental implants lead in most cases to diseases such as osteomyelitis or periodontitis [12]. The administration of antibiotics in this case is carried out for long periods and in large doses, which can lead to the occurrence of systemic toxic effects [13]. Thus, because of these problems, the scientific community continues to research and identify new types of solutions to administer localized antibiotics.

Metallic biomaterials have been extensively studied due to the growing demand. Of these, titanium and its alloys are widely used in the manufacturing process of orthopedic and dental implants due to their suitable features, such as ensuring a good mechanical support for the bone system, along with good biocompatibility and high corrosion resistance [14,15,16]. However, due to its bioinert character, titanium is generally encapsulated in a fibrous tissue after implantation, a process that can eventually lead to implant failure. Thus, in recent years, numerous studies have been carried out on the improvement of metallic implant properties by surface biofunctionalization [17] through various treatments, which can include sandblasting, alkaline treatments, anodizing, or the deposition of bioactive coatings. Among these, we can highlight hydroxyapatite (HAp) due to its low solubility in physiological environments, as it is a bioactive material with a high osseointegration capacity that does not induce inflammatory reactions on direct contact with hard tissues [18,19,20]. Its ability to be osteoconductive propelled it to the top of the list of materials used as coatings [21,22]. Nevertheless, the properties of HAp can be easily improved, as it can replace the Ca^2+^ ions in its structure with various ions such as Sr^2+^, Zn^2+^, Cu^2+^, Fe^2+^, Ag^+^, and Mg^2+^ [23,24].

According to different studies, the doping of hydroxyapatite with strontium led to the improvement of the mechanical properties with the increase in the Sr content [25,26]. It is known that Sr atoms can occupy both the Ca(I) and Ca(II) sites of the apatite lattice, with the substitution at the Ca(I) site being favored for low doping concentrations (1–3.5 at.%) [27,28] and, on the contrary, the substitution at the Ca(II) site being favored for high doping concentrations (>10 at.%) [29]. So far, it has been concluded that the substitution mode depends on the amount of Sr used for doping the HAp.

Strontium has gained attention because it is an element that supports cell proliferation without having toxic effects, and perhaps its most important aspect relies on the fact that it supports the formation of new bone tissue by slowing down osteoclast activity [30]. This important feature makes it a desirable doping element for HAp in the case of implantable medical devices for patients with osteoporosis or osteopenia [31,32,33]. Nowadays, new solutions and approaches for inducing a local antibacterial and/or bactericidal effect are preferred in order to prevent implant-associated infections, rather than the administration of oral or systemic antibiotics. With a broad spectrum of antimicrobial activity, Ag is the most well-known doping element for HAp to improve its antifungal properties [28,34]. The action mechanism of Ag is still being studied. Some research indicates that Ag ions inhibit bacterial cell function by interacting with thiol groups altering DNA [35], while other research sustains that Ag nanoparticles are able to destroy the cell wall by accumulating in the bacterial membrane [36,37].

An important aspect regarding Ag substitution is the amount of Ag used. According to some studies, a high Ag concentration leads to toxicity, affecting the basic cellular functions of all mammalian cells. Considering this, it is necessary to select a proper concentration of Ag so that it has an antibacterial effect and, at the same time, does not induce cytotoxic effects [38,39]. Nevertheless, a quantity of up to 6 at.% favors cell viability and proliferation [40].

In a study carried out by Fielding et al. [41], layers of HAp doped with Ag and/or Sr were obtained by plasma spraying, and antibacterial tests were performed on them using *Pseudomonas aeruginosa*. After 24 h, the layers without Ag showed a high number of live bacteria and very few that had died, while the HAp layers doped with Ag and/or Sr showed a very small number of live cells, indicating the antibacterial efficiency of Ag. Furthermore, Li et al. [42] found that Sr reduced the negative effects of Ag in the case of co-doped HAp, reducing cytotoxicity and favoring cell proliferation. Considering the above-mentioned aspects, it can be emphasized that by co-doping HAp with Sr and Ag, coatings with superior properties can be developed.

In the literature, several methods, such as pulsed laser deposition [43], plasma spraying [44], electrophoretic deposition [45], hydrothermal deposition [46], magnetron sputtering [47], and electrochemical deposition [48], have been used to obtain Hap-based coatings on metallic substrates. Among these, the electrochemical method allows the acquisition of coatings with superior properties due to the parameters that can be easily modified [49]. Also, studies have indicated that titanium has an affinity for the hydroxyl group, thus favoring the formation of HAp coatings through electrochemical deposition [50]. Of all electrochemical deposition techniques currently available, the pulse technique appears to be the most versatile [51].

One of the weak points of the electrochemical method is poor adhesion to metallic substrates [52]. However, the coatings obtained with a relaxation time (t_OFF_) longer than the activation time (t_ON_) through the pulse technique present high crystallinity and adhesion. The concentration of the electrolyte is also an important parameter, as the morphology of the coatings depends on it [53], while the pH of the electrolyte influences the electrochemical reactions and, implicitly, the deposition process [54,55]. Considering all of the above-mentioned aspects, HAp-based coatings deposited on titanium alloys through the pulse technique can be considered a controlled drug release system in the case of implant-related infections.

This study was a follow-up on previous research [56] investigating the influence of the electrolyte’s pH level (pH 4 and pH 5) on the morphology, elemental and phasic composition, and roughness, based on which the layers obtained at pH 5 were selected due to their superior properties compared to the coatings obtained at pH 4. Thus, the aim of this study was to evaluate the electrochemical behavior, biomineralization, and biodegradation capacity in acellular media of cp-Ti and HAp-based coatings undoped and doped with Ag and/or Sr and the cellular response to these substrates.

## 2. Materials and Methods

### 2.1. Electrochemical Deposition of the Coatings

The proposed coatings were electrochemically deposited on commercially pure titanium (cp Ti) metallic substrate cut into discs with a thickness of 2 mm and diameter of 20 mm. The cp-Ti samples were metallographically prepared on silicon carbide paper with different grits (320 ÷ 800), after which they were degreased with isopropyl alcohol and cleaned in acetone for 30 min in an ultrasonic bath (Bandelin, Berlin, Germany). The electrochemical deposition (ED) was performed in a typical three-electrode cell in which the working electrode (WE) was the cp-Ti metallic substrate, the reference electrode (RE) was a saturated calomel electrode (SCE), and the auxiliary electrode (AE) consisted of a platinum sheet. A schematic representation of the obtained coating and surface features can be found in Figure 1.

The coating deposition was carried out with a potentiostat/galvanostat PARSTAT MC equipped with a PMC-1000 module (Princeton Applied Research-AMETEK, Berwyn, PA, USA) using the galvanostatic pulse technique. The electrolyte used for Hap-based coatings was obtained by dissolving the precursor salts Ca(NO_3_)_2_·4H_2_O, NH_4_H_2_PO_4_, AgNO_3_, and Sr(NO_3_)_2_) in ultra-pure water (ASTM I) at different concentrations with a constant Ca/P ratio of 1.67, and the pH was modified to 5 by the dropwise addition of 1 M NaOH. The electrolyte concentrations along with the electrochemical parameters used are presented in Table 1, and a full description of the electrochemical deposition is available in Ref. [56].

### 2.2. Characterization

#### 2.2.1. Coating Thickness

The coating thickness was determined by measuring the level difference at the substrate–coating interface with a stylus profilometer (DEKTAK 150, Veeco Instruments, Plainview, NY, USA).

#### 2.2.2. Chemical Bonds

The Fourier transform infrared spectra were investigated in the 4000–600 cm^−1^ wavenumber range using a FT-IR Jasco 6300 (Jasco, Tokyo, Japan) with the Pike MIRacle ATR sampling accessory (Pike Technologies, Madison, WI, USA).

#### 2.2.3. Wettability

The measurement of the contact angle (CA) was carried out using an Attention Optical Tensiometer Theta Lite 101 (TL) goniometer (Biolin Scientific, Stockholm, Sweden). A Hamilton micro-syringe (Hamilton Company, Reno, NV, USA) was used to drip the liquids on the investigated materials’ surfaces. The angle between the investigated surface and the liquid was measured using the sessile drop method. In order to determine the surface free energy (SFE) of the investigated surfaces, three different liquids were used. Thus, the CA was measured in triplicate for each medium, namely, distilled water (DW) (polar liquid) and ethylene glycol (EG) and toluene (T) (dispersive liquids). Knowing the surface tensions of the liquids, the Fowkes method [57] was used to determine the total free energy of the sample ($\gamma_{S}^{tot}$), consisting of the dispersive component ($\gamma_{S}^{d}$) and the polar component ($\gamma_{S}^{p}$). The experiments were performed in the same conditions, namely, at 24 °C and 37% humidity, in triplicate. The surface tension of all liquids used for surface free energy determination is presented in Table 2 [58,59,60].

In order to correlate the surface energy to the in vitro behavior of the investigated samples, the contact angles with simulated body fluid (SBF), which resembles human plasma, were measured. The chemical composition of the SBF medium is presented in Table 3.

#### 2.2.4. Electrochemical Behavior in Simulated Body Fluid

The electrochemical behavior was determined through the polarization technique in SBF. The electrochemical tests were carried out using a PARSTAT 4000 potentiostat/galvanostat (Princeton Applied Research—Ametek, Oak Ridge, TN, USA), to which a low-current interface module (Princeton Applied Research—Ametek, Oak Ridge, TN, USA) was coupled.

A standard electrochemical cell was used to perform the tests, consisting of a saturated calomel electrode (SCE) as the reference electrode, a platinum electrode as the counter electrode, and a working electrode (Teflon holder) in which the investigated samples were introduced. All electrochemical measurements were carried out at human body temperature (37 ± 0.5 °C), held constant by a heated circulating bath (CW-05G, Jeio Tech, Yuseong-gu, Daejeon, Republic of Korea). Prior to the potentiodynamic measurements, the open circuit potential (E_OC_) was monitored continuously for 3 h immediately after the samples were immersed in the electrolyte, for a potential-stabilizing immersion period of the working electrode in the testing media.

The Tafel plots were measured from −200 mV (vs. E_OC_) to +200 mV (vs. E_OC_) with a scanning rate of 1 mV/s. With the help of these experiments, the corrosion potential (E_corr_) and the corrosion current density (i_corr_) were estimated by Tafel plot extrapolation. 

The polarization resistance (Rp) was computed using the Stern–Geary equation [61] according to the ASTM G59-97 standard (reapproved 2020) [62]:(1)$$R_{p}=\frac{1}{2.3}\cdot \frac{\beta_{a}\cdot \left|\beta_{c}\right|}{\beta_{a}+\left|\beta_{c}\right|}\cdot \frac{1}{i_{\mathrm{corr}}}$$
where *β_a_* and *β_c_* are the anodic and cathodic Tafel slopes, respectively, and i_corr_ is the corrosion current density. The protection ability of the coatings against the SBF medium was assessed using the protective efficiency parameter (P_e_) determined with the following formula [63]:(2)$$P_{e}=\left(1-\frac{i_{\mathrm{corr},\;\mathrm{coating}}}{i_{\mathrm{corr},\;\mathrm{substrate}}}\right)\times 100$$
where i_corr, coatings_ and i_corr, substrate_ are the corrosion current densities of the coating and of the substrate, respectively.

#### 2.2.5. In Vitro Bioactivity Assays in Acellular Media 

In order to evaluate the in vitro behavior, such as the biomineralization and biodegradability of the coatings, the cp-Ti samples and HAp-based coating were immersed in SBF and phosphate-buffered solution (PBS) for 1, 3, 7, 14, and 21 days at human body temperature (37 ± 0.5 °C) in an incubator (Memmert IF 55). Both media are commonly used to predict material behavior in vivo. The chemical composition of PBS is presented In Table 3.

The two media were changed every 3 days as the concentration of ions decreased due to the chemical changes in the material, but also to prevent the growth of microorganisms. 

To investigate their evolution, the samples were extracted at different time points (1, 3, 7, 14, and 21 days) and weighed with an analytical balance (Kern, ALT 100-5AM, Balingen, Germany) with an accuracy of 0.01 mg. The samples were removed from the testing media at predetermined time intervals, rinsed with distilled water, dried in a hot air jet, and then stored in a desiccator until weighing.

### 2.3. In Vitro Biocompatibility Assessment

#### 2.3.1. Cell Seeding and Culture

In vitro assays for the biocompatibility testing were performed by direct cell–surface contact studies of the cp-Ti and the HAp-based coatings using the MC3T3-E1 murine preosteoblast cell line (ATCC^®^ CCL-2593 ™, American Type Culture Collection, Manassas, VI, USA). The preosteoblasts were seeded on the samples sterilized by UV exposure at an initial cell density of 15 × 10^3^ cells/cm^2^ in all experiments, except for osteogenic differentiation studies, where a cell density of 40 × 10^3^ cells/cm^2^ was used. Afterwards, the samples were incubated for different periods in Dulbecco’s modified Eagle medium (DMEM) supplemented with 10% fetal bovine serum (FBS) and 1% penicillin/streptomycin/amphotericin at 37 °C in a humidified atmosphere of CO_2_. The preosteoblast differentiation studies were conducted in pro-osteogenic culture conditions (cell incubation in DMEM containing 50 μg/mL ascorbic acid (Sigma-Aldrich, Burlington, MA, USA) and 5 mM beta-glycerophosphate (Sigma-Aldrich).

#### 2.3.2. Cell Viability Study—LIVE/DEAD Assay

A LIVE/DEAD Viability/Cytotoxicity kit (L-3224, Invitrogen, Carlsbad, CA, USA) was used to assess the viability of MC3T3-E1 cells grown in contact with the tested materials. This kit involved the simultaneous action of a Calcein acetoxymethyl ester (AM) and ethidium homodimer (EthD-1). Thus, after 1 and 3 days, the samples were washed with PBS, treated with a solution containing Calcein AM:EthD-1 (2 M:4 μM) and washed again with PBS, as previously reported [64]. After that, the cell-populated materials were visualized under an Olympus IX71 fluorescence microscope. Representative fields were captured using the Cell F imaging system.

#### 2.3.3. CCK-8 Cell Proliferation Assay

The number of metabolically active viable cells in contact with the tested surfaces was quantified by means of a Cell Counting Kit-8 (CCK-8, Sigma-Aldrich Co., St. Louis, MO, USA), in accordance with the manufacturer’s instructions [65].

#### 2.3.4. MC3T3-E1 Preosteoblast Morphology—Fluorescent Labeling of Actin Filaments

The fluorescent labeling of actin filaments was performed to highlight the morphological changes in the MC3T3-E1 preosteoblasts induced by the analyzed HAp-based coatings compared to cp-Ti. Thus, the cells adhered to these materials were fixed at different time intervals, respectively, 1 and 3 days after seeding, with a cold solution of 4% paraformaldehyde in PBS for 20 min and then processed to evince actin cytoskeleton (Alexa Fluor™ 488 phalloidin staining, Termo Fisher Scientific, Waltham, MA, USA) and nuclei (DNA labeling with, 4`6-diamidino-2-phenylindole (DAPI)) [66]. The tested cell-populated samples were visualized under an inverted fluorescence microscope (Olympus IX71, Olympus, Tokyo, Japan), and representative images were captured using the Cell F image acquisition system (v5.0).

### 2.4. The Effect of Test Samples on Osteogenic Differentiation

#### 2.4.1. Determination of Alkaline Phosphatase Activity

The alkaline phosphatase activity (ALP) was determined after 7 days and 14 days of culture on cell lysates using a commercial kit (Alkaline Phosphatase Activity Colorimetric Assay Kit, BioVision; Milpitas, CA, USA), as presented in a previous paper [67]. Alkaline phosphatase activity was calculated according to the formula:ALP (U/)mL = A/V/T(3)
where A represents the amount of p-nitrophenol (pNP) expressed by the samples (in μmol), V is the volume of lysate used in the reaction (in mL), and T represents the reaction time (in minutes). 

To avoid variations due to different protein concentrations, all samples were reported to the protein concentration which was previously determined using the Bradford method. Thus, the concentrations of the reaction product (p-nitrophenol) were normalized to 1 μg protein.

#### 2.4.2. Determination of the Level of Synthesized and Secreted Collagen

The level of collagen synthesized and extracellularly secreted by the MC3T3-E1 cells grown in contact with all analyzed materials for 14 and 21 days was determined by staining with a Sirius Red solution, according to a previously published protocol [68]. 

#### 2.4.3. Quantitative Evaluation of Extracellular Matrix Mineralization 

For quantifying the extracellular matrix mineralization (ECM), the cell monolayer formed on the materials’ surfaces was processed as previously reported [67]. Briefly, after washing with PBS and fixation with 10% formaldehyde solution, the samples were stained with Alizarin red solution. The stained calcium deposits were quantified by solubilizing them with 5% perchloric acid, and the absorbance was read at 405 nm. To eliminate the specific colorations of HAp coatings, the coloring of these samples without cells was also performed, and the recorded OD value was subtracted from the OD values obtained on the corresponding sample populated with cells.

## 3. Results and Discussions

### 3.1. Coating Thickness

In terms of coating thickness, the analyzed profiles (Figure 2) showed that the smallest coating thickness of 11.8 μm (±0.3) was obtained for the undoped hydroxyapatite codified with H. 

The addition of Sr into the HAp structure led to the highest thickness registered, with a measured value of 13.3 μm (±0.5), while the addition of Ag led to a medium thickness, registering a value of 12.7 μm (±0.5). When both elements were added to the HAp structure, a decrement in the coating thickness was observed, reaching a value of 12.2 μm (±0.6), which was smaller than those obtained for the H-Ag and H-Sr samples. Thus, based on the information gathered on the coating thickness, it could be observed that the values were between 11.8 μm and 13.3 μm.

Given that the electrochemical parameters involved in the deposition process were kept constant for all coatings, it could be assumed that the coating thickness difference was due to the addition of the doping elements, which influenced the nucleation and crystal growth rate of the HAp. 

It was shown that within the first 15 min, the coatings comprised plate-like crystals, which tended to elongate in the *c*-axis direction and grow perpendicular on the substrate, forming ribbon-like crystals [69,70]. Thus, it can be said that the crystal growth of CaP species is a time-dependent process, and the deposition mechanism is related to the migration and ionic concentration of Ca^2+^, PO_4_^3−^, and OH^−^ in the vicinity of the cathode, which is dependent on the applied current density over the working electrode [69]. In terms of the electrochemical configuration, it is known that in a three-electrode setup, the applied current density is maintained with respect to the reference [71], allowing good control over the coating thickness. The deposition rate was computed as a ratio between the coating thickness and the active time of deposition (t_ON_) expressed in μm/min (Figure 2b).

As can be observed from Figure 2b, the smallest deposition rate was registered for the undoped HAp (H) coatings (0.79 μm/min), closely followed by the co-doped HAp, H-Sr-Ag (0.81 μm/min), with an increment of only 3% in the deposition rate. The highest deposition rate was registered for the H-Sr coatings (0.89 μm/min), which in comparison with the undoped HAp registered an increment in the deposition rate of 11%, while the H-Ag (0.85 μm/min) coatings had an increment of 7%, indicating that each element had a different impact on the deposition rate of the coatings.

It is known that the four Ca(I) sites within the hexagonal structure of HAp are tightly bonded to six oxygen atoms and less strongly bonded to three oxygen atoms with a mean Ca(I)–O bond length of 0.255 nm, whereas the six Ca(II) sites are surrounded by seven oxygen atoms with a mean Ca(II)–O bond length of 0.245 nm. The shorter distances between the Ca(II) sites and their alignment in columns make them energetically favorable for cations of a smaller size compared to Ca atoms, whereas larger cations should be accommodated in the Ca(I) sites [72,73,74]. It is known that Sr atoms can occupy both the Ca(I) and Ca(II) sites of the apatite lattice, with substitution at the Ca(II) sites favored for high doping concentrations and, on the contrary, substitution at the Ca(I) site favored for low doping concentrations (1.0–3.5 at.%) [28].

Our previous results [56], which showed that the Sr content reached 3.5 at.%, suggested that the Sr ions replaced part of the Ca(I) site. Similarly, Ag^+^ ions substituted Ca^2+^ ions within the HAp lattice, preferring the Ca(I) site, which led to an increment in the lattice parameters due to the greater ionic radius (0.128 nm vs. 0.099 nm) [28]. This finding agrees with previously published results [56] on these coatings showing that after the addition of Sr and/or Ag, the lattice parameter increased in comparison to that of undoped Hap, and some diffraction maxima shifted towards smaller angles.

The fact that both doping elements, Sr^2+^ (0.120 nm) and Ag^+^ (0.128 nm), presented a higher ionic radius than Ca^2+^ (0.099 nm), along with their tendency to substitute Ca(I) within the HAp lattice, indicates that when both elements are present in HAp, there is competition among them to substitute the Ca(I) site, leading to the lower coating thickness of the H-Sr-Ag sample in comparison with the H-Sr and H-Ag samples.

### 3.2. Chemical Bonds

To confirm the presence of the phosphate functional group in the Hap-based coatings, FTIR analysis was carried out, and the results are shown in Figure 3. The bands found between 1026 and 1118 cm^−1^ arose from the asymmetric stretching vibration mode of PO_4_^3−^ (ν_3_), the ~964 cm^−1^ band arose from the symmetric stretching vibration modes of PO_4_^3−^ (ν_1_), and the ~640 and 601 cm^−1^ bands arose from the asymmetric bending vibration of PO_4_^3−^ (ν_4_) [75,76], indicating that the main vibrations of the phosphate functional groups were present in the obtained HAp-based coatings.

Two types of OH^−^ could be identified in the IR spectra, namely, the stretching mode at ~3570 cm^−1^ owing to the existence of an organized H_2_O structure in the HAp, and the bending mode at ~1634 cm^−1^ belonging to the H_2_O molecules [77,78]. In the case of HAp, two bands due to OH^−^ ions are typically detectable at ~632 cm^−1^ (as a shoulder of the ν_4_(PO_4_^3−^)) and ~3572 cm^−1^ (narrow peak), which was also the case for the proposed coatings (Figure 3).

In principle, the lack of OH^−^ in bone apatite could be attributed to either the presence of CO_3_^2−^, through the direct displacement of two OH^−^ ions by one CO_3_^2−^ ion in the channel site, also known as A-type substitution, or to the demands of charge balance in the so-called B-type substitution [79]. Thus, it appears that the carbonate ion can substitute for both OH^−^ in the *c*-axis channel of apatite (type-A carbonate) and the phosphate group (type-B carbonate). It is understood that type-A carbonate is characterized by a doublet band at approx. 1545 and 1450 cm^−1^ (asymmetric stretch vibration, ν_3_) and a singlet band at approx. 878 cm^−1^ (out-of-plane bending vibration, ν_2_), whereas type-B carbonate has these bands at about 1455, 1410, and 873 cm^−1^, respectively [80].

However, for nonstoichiometric HAp obtained by wet methods at moderate temperatures, these OH^−^ bands are often difficult to detect by IR spectroscopy, due to the lower degree of crystallinity that tends to enlarge the vibrational bands, covering weak OH^−^ signals [81]. Since the coatings were prepared by the electrochemical deposition process, which can be categorized as a wet method, the resulting coatings were generally associated with water molecules. Thus, the broad band between 3600 cm^−1^ and 3000 cm^−1^ was attributed to the water adsorbed at the surface and the characteristic stretching vibrational mode of the structural OH^−^ group, while the band at 1630 cm^−1^ derived from the bending mode of the H_2_O molecules [82,83].

The peak identified at around 870 cm^−1^ in the IR spectra could be attributed to CO_3_^2−^ or HPO_4_^2−^, since a part of the PO_4_^3−^ from the HAp structure could be substituted by this type of ion [80]. It is known that a part of the PO_4_^3−^ can be replaced by CO_3_^2−^, probably formed by the dissolution of CO_2_ gas during the electrochemical deposition process [77,78], but since no carbon was detected by energy dispersive X-ray spectroscopy (EDS, results not shown), it was assumed that the peak detected at ~870 cm^−1^ could be attributed to the HPO_4_^2−^ ion.

According to the literature [81], a thin band at 875 cm^−1^ is also detected for HPO_4_^2−^-bearing apatite due to P-OH stretching. However, if the apatite is partly carbonated (which is noticeable, for instance, from a broad ν_3_(CO_3_^2−^) absorption band in the region 1350–1570 cm^−1^), then the HPO_4_^2−^ band at 875 cm^−1^ becomes almost superimposed on the ν_2_(CO_3_^2−^) band (around 872 cm^−1^ for B-type carbonate ions), making it difficult to exploit [81]. Because no other bands attributed to the CO_3_^2−^ vibration modes were identified in the obtained spectra (Figure 3), it could be assumed that the peak identified at ~870 corresponded to the HPO_4_^2−^ ion.

Therefore, the FTIR analysis indicated that all coatings presented an HAp phase, and all spectra presented similar patterns with some differences in terms of intensities depending on the coating type. According to the investigations presented in our previous research [56], the coatings mostly consisted of the HAp phase according to ICDD #09-0432, with a small amount of monetite as the secondary phase, as demonstrated by the X-ray diffraction analysis, and a (Ca + M)/P ratio (where M = Sr, Ag, and Sr + Ag) with a value smaller than that of the stoichiometric HAp at 1.67. This, corroborated by the presence of HPO_4_^2−^ identified by the FTIR analysis, indicated that the proposed coatings could be classified as calcium-deficient hydroxyapatite.

### 3.3. Contact Angle and Surface Free Energy

Conventionally, the contact angle (CA) can range between 0° and 180°, as a measure of the ability of a liquid to wet a solid surface. If the CA is lower than 90°, the surfaces are designated as hydrophilic, while at values above 90°, the surfaces are designated as hydrophobic. Solid surfaces can also be superhydrophilic or superhydrophobic when the CA is very close to 0° or above 150°, respectively [84].

The surface energy of an implant can be measured indirectly by the liquid–solid contact angle, and it is related to the wettability, which is a property known to impact the biological response to an implant during the initial stages of wound healing and during the cascade of events that occurs during osseointegration, facilitating bone integration [85]. It is known that the wetting degree is correlated with the surface free energy, and both depend on the surface roughness [86,87]. Such surface supports the adhesion of proteins, and it is a crucial parameter on which the bone–implant contact depends [88,89].

Table 4 presents the values of the contact angle with distilled water, ethylene glycol, and toluene obtained for the cp-Ti substrate uncoated and coated with undoped and doped HAp layers, used to calculate the surface energy. It was noted that irrespective of the liquids used, all materials had a hydrophilic character, with values of the CA smaller than 90°. Comparing the CA values obtained for the HAp-based coatings, which were between 7.5° and 10.5°, with those of the Ti substrates, at ~68°, it could be observed that the values of the latter were significantly higher, by approximately one order of magnitude.

Based on the values presented in Table 4, the surface energy was calculated and is presented in Figure 4, along with the contact angle obtained while using SBF for all investigated materials.

The smallest SFE was obtained for the Ti substrate, with a value of ~35 mN/m, which was in agreement with other studies [90,91], while the deposition of the HAp-based coatings on the Ti substrate led to an increment in the SFE, with the highest value reached for the H-Sr-Ag sample, followed by the H-Sr, H-Ag, and H samples. The SFE is influenced by surface chemistry and surface roughness, which are two different parameters that do not influence each other (modifying the surface chemistry does not affect the surface roughness, and the reciprocal is also valid).

Regarding the surface chemistry, it is known that the polar component is characterized by strong bonds between molecules, such as hydrogen, ionic, and/or covalent bonds [92], while the dispersive component is characterized by weak bonds (van-der Waals forces). Thus, based on this and considering the information gathered from the FTIR analysis, it can be said that the increment in the SFE was due to the presence of OH^−^ and PO_4_^3−^ functional groups identified in all HAp-based coatings [93]. It is known that hydrophilic surfaces are characterized by a higher polar component and SFE [94,95]. According to this criterion, Figure 4a demonstrates that the polar component significantly prevailed over the dispersive components of the SFE. In the case of the dispersive component of the SFE, it was observed that all HAp coatings presented approx. the same value of 2 mN/m, while the polar component registered different values in all coatings, which may have been due to the coatings’ roughness and/or the presence of the doping elements.

Compared to a smoother surface, a rougher material has a larger total surface area due to the presence of small valleys/depressions, which offer a larger wetting area, and thus the wettability increases. In this case, the high roughness was rendered by the morphology of the layers specific to electrochemical deposits [96].

The values of the average surface roughness (Ra) parameter published in our previous paper [56] showed that all coatings had an Ra between 390 and 402 nm, except for the H-Sr-Ag coatings, which had an Ra of 680 nm. Thus, one can see that even though the SFE slightly increased in accordance with the doping element, it may not be the only criterion to consider when discussing the wettability of this type of surface.

When the doping element is considered, one can see that even though no major differences were noted among the SFE values obtained for the HAp-doped coatings, it was observed that the addition of Sr had a slightly higher impact than the addition of Ag on the wettability degree. Thus, all HAp-doped coatings had a higher SFE than the undoped HAp, with the highest SFE reached when both doping elements were present within the HAp coating.

Regarding the contact angles obtained for the investigated materials when SBF media were used, it can be said that the substrates coated with undoped HAp and HAp doped with Sr and/or Ag had a strong hydrophilic character, identified by contact angles smaller than 15°. Compared to the values registered for the pure Ti substrate, the coatings registered smaller values, highlighting the beneficial effect of the coatings on the wettability properties.

In terms of the doping element, the results revealed that the addition of Sr led to a slight increment in the contact angle, from a value of 8.86°, obtained for the undoped HAp, to a value of 10.32°. On the other hand, the addition of Ag led to a contact angle of 8.6°, which was very close to that obtained for the undoped HAp, 8.86°, in accordance with the literature [97]. When both doping elements, Sr and Ag, were added to the HAp coatings, the contact angle had an identical value (8.86°), suggesting that the two doping elements complemented each other.

Given the values of the CA when SBF media were used, it can be said that the coatings exhibited a superhydrophilic character. Superhydrophilic surfaces (contact angle <10°) have been extensively studied, as they improve cell proliferation and attachment and help increase osseointegration capacity [98,99].

Thus, based on the obtained results, it can be emphasized that the coatings’ morphologies correlated with the roughness and contact angle [100], and the improvements in the surface features of the HAp-based coatings should favor cell adhesion and proliferation [101].

### 3.4. Electrochemical Behavior

The most well-known form of corrosion encountered on the surface of orthopedic and dental implants is pitting corrosion, which is induced by the chlorine ions (Cl^−^) present in the fluids of the human body [102,103]. The excellent corrosion resistance of Ti and its alloys is a direct result of Ti’s affinity for oxygen [104], which leads to the formation of a passive layer 1.5–10 nm in thickness on the surface [105], preventing the possible release of metallic ions into the human blood stream [106]. This passive layer is exposed to repeated partial dissolution and reprecipitation in aqueous solutions, and any disruption of this balance, namely, under abnormal cyclic loads, micromotions, acidic conditions, and so on, can lead to film breakdown, causing metal ion release [107,108]. Thus, under these circumstances, the deposition of coatings on Ti-based metallic surfaces may come as an additional barrier between the possible release of metallic ions and the surrounding tissues and environment [109].

The variations of the open-circuit potential (OCP) and the Tafel plots corresponding to the cp-Ti and HAp-based coatings are presented in Figure 5. Based on the obtained results, the main electrochemical parameters were extracted and can be found in Table 5.

A more electropositive value of the open-circuit potential (E_OC_) denotes a more noble character from the electrochemical point of view. On the other hand, a more electropositive corrosion potential (E_corr_) and a low value of corrosion current density (i_corr_) denote better electrochemical behavior. Thus, the most electropositive E_OC_ was recorded for the H-Ag (−37.88 mV), followed by the H-Sr-Ag (−46.93 mV), H-Sr (−154.12 mV), cp-Ti (−170.38 mV), and H (−171.07 mV), respectively. Knowing that silver has a high positive value in the electrochemical series of 0.8 V, it can be assumed that this increment in the open-circuit potential for the H-Ag and H-Sr-Ag samples may have been due to the presence of silver ions in the hydroxyapatite structure.

The corrosion potential of H-Ag had the most electropositive value (−63.20 mV), followed by H-Sr-Ag, which recorded a value of −84.42 mV. There were no significant differences between the H and H-Sr coatings, whereas their corrosion potentials were −166.24 mV and −161.07 mV, respectively. The cp-Ti displayed a corrosion potential of −129.79 mV, a higher value than in the case of the H and H-Sr coatings, which was most likely due to the increased oxide layer on the surface of the substrate.

If we take into consideration the corrosion current density (i_corr_) parameter, it can be noted that the smallest value was registered for the H coatings (18.07 nA/cm^2^). The addition of Sr (H-Sr) within the HAp coatings led to a slight increment in the i_corr_ to 26.27 nA/cm^2^, while the coatings doped with Ag (H-Ag) registered the highest value of this parameter (52.63 nA/cm^2^), in agreement with the literature [110]. However, by co-doping the HAp with silver and strontium, the value of i_corr_ reached 48.63 nA/cm^2^, which fell in between those obtained for the H-Ag and H-Sr samples.

Based on the values obtained for the E_corr_ and i_corr_ parameters, it could be assumed that during the immersion in the testing media, dissolution processes of different doping elements as well as calcium phosphate components could occur [75,111].

It is known that a high polarization resistance (Rp) indicates the good electrochemical behavior of a material, and a low value of this parameter shows poor electromechanical behavior. From this point of view, it was observed that the H coating also had the highest value of this parameter, thus demonstrating better electrochemical behavior than the other samples. At the same time, there was a significant difference between the Rp values of the tested samples, due to the hydroxyapatite and the doping element.

The addition of the doping element led to a decrement in the polarization resistance in comparison with the undoped HAp, in the following order: H > H-Sr > H-Sr-Ag > H-Ag. Nonetheless, it could be observed that irrespective of the coating type, all Rp values were higher than those reached for the cp-Ti sample, highlighting the beneficial effect of the proposed coatings.

The evolution of the Rp electrochemical parameter resembled the i_corr_ trend, which was most likely due to the metallic character of Ag and may indicate that Ag ions can be released in the environment to inhibit the formation of a bacterial film.

In terms of the protective efficiency (P_e_) against the aggressive attack of the SBF medium, the undoped HAp coatings had the highest efficiency of 78.4%, closely followed by the H-Sr coatings with a value of 68.5%. For the H-Ag coatings, the addition of Ag into the HAp structure reduced the efficiency by half in comparison with HAp, reaching 37%, representing the smallest of all obtained values. Nonetheless, in comparison with the H-Ag coatings, the H-Sr-Ag coatings registered a small enhancement in the efficiency of approx. 5%, reaching a value of 41.8%.

Thus, after carefully analyzing the electrochemical parameters presented in Table 5, it can be emphasized that all proposed coatings enhanced the electrochemical behavior of the Ti substrate. Nonetheless, the best electrochemical behavior was demonstrated by the H coatings, which obtained the lowest corrosion current density and highest polarization resistance and protective efficiency, being closely followed by the H-Sr coatings.

Even though the coatings doped with silver, namely, H-Ag and H-Sr-Ag, indicated poorer electrochemical behavior in comparison with the H and H-Sr samples, this did not mean that they possessed weak properties. This was most likely due to the metallic character of Ag, which could confirm the fact that, in this case, Ag was not completely substituted in the structure of the hydroxyapatite and existed as nanoparticles.

### 3.5. Evaluation of In Vitro Behavior in SBF and PBS

Immersion in simulated body fluid (SBF) can be used to predict the biomineralization capacity and osseointegration of an implant. During immersion in SBF, new calcium phosphates can precipitate on the surface of the coatings, allowing one to predict their behavior in vivo [110].

The mass evolution of cp-Ti and HAp-based coatings immersed in SBF is presented in Figure 6. In comparison with cp-Ti, all coatings showed a mass increment even after 1 day of immersion in SBF. It can be noted that H-Sr-Ag displayed the highest mass gain after 21 days, while H-Ag coatings registered the lowest value of all investigated coatings. This correlated strongly with the electrochemical results, where H-Ag coatings exhibited the poorest behavior.

If we also consider the CA obtained when SBF media were used, it can be noted that the H and H-Sr-Ag coatings registered similar values (CA_SBF_ = 8.86°), while in the case of H-Sr and H-Ag, the CA_SBF_ values were 10.32° and 8.60°, respectively. Even though the differences in the CA in SBF among the samples were not that high, it can be observed that the coatings with a value of ~8.9° registered the highest mass gains, while the coatings doped with one element, namely, H-Sr and H-Ag, registered smaller values.

The H-Sr-Ag coatings registered values close to those of the H-Sr and H-Ag samples, irrespective of the immersion period. However, in all cases, the mass evolution increased after each immersion period. Based on these results, it could be assumed that the increment in the immersion period in SBF favored the precipitation of newly formed apatite, regardless of the HAp coating type [112].

According to the literature, Sr in HAp promotes bone binding ability [113,114], while the addition of Ag tends to delay the formation of new apatite crystals [115]. In the current scenario, it could be observed that when both elements were added, the mass of the newly formed apatite reached the highest value, improving the biomineralization ability of the HAp-based coatings [113].

The immersion assays in PBS revealed that the mass loss increased for each HAp-based coating, thus demonstrating that the addition of doping elements improved the degradation capacity of the HAp (Figure 7). From the chemical point of view, the cp-Ti surface was stable, and even though it was not susceptible to degradation, a slight increment in its mass was noted, which could be attributed to the formation of small salt deposits on its surface. Thus, the highest decrease in mass was noted for the H-Sr-Ag samples, at −1.12 mg, followed by H-Ag with −1.08 mg and H-Sr with 1.05 mg, while the H coatings showed a decrease of −0.87 mg after 21 days of immersion. A possible explanation for the dissolution rate registered in PBS for the HAp-based coatings could be attributed to the presence of an amorphous phase in the HAp structure [56]. Another possible explanation could be that the coatings’ morphology, comprising long ribbon-like crystals that grew perpendicular to the substrate, led to the development of small voids inside the coatings, which were favorable for the vascularization of the implant. Thus, it could be assumed that the differences between the dimensions of these ribbons induced different degrees of porosity for each type of coating. Diez-Escudero et al. [116] also found that the porosity had a significant influence on the degradation rate of the coatings.

It cannot be neglected that HAp presents a low solubility, which could be controlled by adding different biocompatible elements, such as those used in the current study (Ag and Sr), into the HAp structure.

The optimum concentration of Sr in biomaterials has been the subject of much debate, without a consensus being reached. Several concentrations of Sr aiming to obtain optimal bone regeneration can be found in the literature [117,118,119]. A Sr^2+^ concentration of 100–140 ppb was not sufficient to promote cell proliferation [120], whereas at concentrations between 210 and 21,000 ppb [121], some positive effects were reported. In the case of post-menopausal osteoporotic women treated with 2 g/day of Sr ranelate, which delivered 680 mg of elemental strontium [122], 0.05 wt.% and 8.7 ppm of Sr were found in the hard bone tissue and blood [73,123], representing a much lower amount in comparison with the in vitro/vivo assays mentioned above. Nonetheless, the usage of Sr ranelate may cause headaches, nausea, gastrointestinal discomfort, and some high-risk pathologies such as venous thromboembolism or myocardial infraction [124].

It is known that the addition of Sr as a doping element influences the morphology, apatite structure, and crystallinity of CaP ceramics [125], which further impact the coatings solubility, in vitro cell behavior, and in vivo bone formation and remodeling [126,127]. In bone tissue, Sr^2+^ is found at a concentration of 0.008–0.010 at.%, whereas the concentration in actively metabolizing bone can increase to between 3 and 7 at.%, indicating the importance of Sr^2+^ within the bone remodeling process [72,127]. Since bone formation is stimulated and enhanced by low amounts of Sr, but mineralization is reduced at higher concentrations [119], it can be said that bone formation is dependent on the Sr dosage.

The theoretical limit of Ag^+^ substituting Ca^2+^ is 20% [128], but the practical limit is much lower than this. At a low concentration, the Ag^+^ ions would replace some of the Ca^2+^ ions in the HAp structure without changing the crystal structure [129], but unfortunately, the Ag-doped HAp material could also be toxic to cells when the Ag concentration exceeds 6 wt.% [130]. Moreover, a too high Ag concentration can result in the formation of amorphous HAp and a performance decrement [131]. Also, high levels of Ag in the human body are associated with leukopenia and liver and kidney damage, while long-term exposure to silver in human blood can cause argyria [132,133].

In terms of in vitro cell behavior, the results have indicated that 1–4 wt.% Ag has no cytotoxic effect, and 1–2 wt.% is efficient against *Staphylococcus aureus* (*S. aureus*) [133]. The ionic release of Ag^+^ is controlled by water diffusion in the surface pores of coating films; thus, it is essential to incorporate the minimum amount to reduce bacterial adhesion on biomaterials as well as to minimize host tissue toxicity [38].

For the preparation of the HAp-based coatings doped with Sr and/or Ag, the research design took into consideration the concentration threshold of each doping element, and, as already proven [56], the concentrations did not exceed the above-mentioned limits, such as 6 wt.% for Ag.

According to Williams [134], there are two fundamental types of mechanism that could be involved in bone-inducing bioactive materials: (1) the metal ion influence, for which calcium is likely to dominate, though other elements can control the balance between osteoblast and osteoclast behavior or the differentiation of stem cells into osteoblasts, and (2) the material topography, which are governed by the interplay between stresses at or near the interface and are generated by endogenous or exogenous factors.

The addition of ions with a larger radius than Ca^2+^, such as Ag^+^ and Sr^2+^, led to an increment in the lattice parameters that could induce internal stress and strain and thus the deformation of the HAp crystallographic structure. In terms of bioactivity assays, this could be translated into small structural defects that allowed ionic exchange between the HAp-based coating and the environment, hence favoring the precipitation of a new layer of apatite on the material surface in SBF and degradation in PBS.

Based on the results obtained from the immersion assays and their correlation with those related to the electrochemical behavior, especially the i_corr_ evolution, it could be assumed that the addition of Ag and Sr favored the solubility of HAp.

### 3.6. MC3T3-E1 Cell Viability and Proliferation

The capacity of the HAp coatings to sustain cell viability and proliferation was evaluated by combining the results of the LIVE/DEAD cell viability and CCK-8 colorimetric assays. Following preosteoblast staining with the LIVE/DEAD Viability/Cytotoxicity Kit, it was noted that all surfaces were mostly populated with preosteoblasts, marked by green fluorescence (living cells), indicating the absence of cytotoxic effects (Figure 8).

As can be seen, when in contact with the HAp-based coatings, the MC3T3-E1 cells underwent some morphological changes, presenting an elongated aspect and an increment in their density from 1 day to 3 days of culture, suggesting that these surfaces supported cell proliferation. As expected, the highest cell density values were recorded for the Ti substrate at both incubation times, which could be explained through the presence of the stable and inert oxide layer that spontaneously forms when the surface is exposed to oxidizing media [135]. Another explanation could be the contact angle recorded for cp-Ti, which, according to the literature, was the optimal value to support cell viability and proliferation [136].

Note that in the case of the H-Ag and H-Sr-Ag coatings, a small number of red fluorescent cells stained with the ethidium homodimer, highlighting the nuclear material of the dead cells with compromised plasma membranes, was evinced. Since dead cells were noticed only on the coatings to which Ag was added, it could be assumed that this behavior was due to the presence of metallic Ag found in the flower-like agglomerations [56], which could induce respiratory chain disturbances, DNA denaturation/condensation [137], and membrane structural changes in both prokaryotic and eukaryotic cells [138]. Thus, it could be assumed that the cell death may have been a direct cause of the metallic Ag present in the H-Ag and H-Sr-Ag coatings, which when in contact with the cells and culture media led to the release of a very high amount of Ag ions from the coating, inducing localized cell death.

According to [42], Ag in ionic form is cytotoxic if it is found in amounts greater than 10 wt.%. In the case of the coatings that were developed in the present study, H-Ag and H-Sr-Ag registered lower amounts (3.02 and 2.4 wt.%, respectively). Also, Chambard et al. [139] found that the presence of Ag nanoparticles could be beneficial if they were uniformly distributed on the surface of the coatings and if they were small. Our Ag-doped coatings had a ribbon-like morphology coupled with flower-like agglomerations where metallic Ag could be encountered; however, their distribution was not uniform.

Considering the results of the CCK-8 assay, if we compare the coatings, it can be observed that after 1 day of incubation, all coatings, irrespective of the doping element, presented very close values, with no significant differences (Figure 9). After 3 days of cell incubation, differences between the HAp-based coatings could be observed, as pointed below. The addition of Sr into the HAp structure enhanced the cell proliferation rate, as can be observed in both Figure 8 and Figure 9, showing a higher density of preosteoblasts and a slight increase in the optical density (OD) values, respectively, when compared with the H and H-Ag samples. Moreover, the presence of Sr within the H-Sr and H-Sr-Ag samples did not significantly affect the cell proliferative capacity in comparison with the Ti substrate, while H and H-Ag elicited a significantly reduced proliferation rate (*p* < 0.01 vs. Ti) in accordance with the results of the LIVE/DEAD assay (Figure 8). The obtained data were in line with those of a previous paper [126] revealing the stimulatory effects of Sr in Sr-HAp materials on the proliferation of human MG-63 osteoblasts.

### 3.7. Cell Morphology

Following the double-labeling experiments for cytoskeletal actin (using phalloidin coupled with Alexa Fluor 488) and nuclei (using DAPI), the morphological differences between cells grown in contact with cp-Ti and HAp-based coatings were highlighted (Figure 10).

Thus, on cp-Ti, the preosteoblasts adopted an elongated polygonal conformation with stress fibers distributed along the cell body, while on the HAp-based coatings, they showed a filiform, dendritic conformation after 1 day of incubation, which transformed into a predominantly cuboidal conformation after 3 days of incubation.

Furthermore, on these coatings the cells exhibited fine, elongated cytoplasmic extensions that established intercellular contacts. Likewise, no stress fibers were obvious, suggesting a lower number of focal adhesions at the cell–material interface and, implicitly, weaker cell–material contact, most likely due to the poor adhesion of the serum proteins from the culture media.

### 3.8. Cell Differentiation

Osteoblasts are bone-forming cells, and their main role is to support bone reconstruction, which is a part of the osseointegration process. The differentiation of osteoblasts is a stage in the osseointegration process, which in turn is carried out in several stages [140]. At the end of their functional program, the osteoblasts can become: (i) mature bone cells (osteocytes), which are the most abundant bone-derived cells with the longest lifespan [141], specialized at maintaining the mineral concentration of the extracellular matrix [142]; (ii) cells that cover the hard tissue (bone-lining cells), which are considered to play an important part in coupling the processes of bone resorption and bone formation, or can undergo apoptosis [143].

In the mineralization process, the intracellular alkaline phosphatase (ALP) plays an important role in the formation of hydroxyapatite by producing its constitutive elements, PO_4_^3−^ and Ca^2+^ [144]. However, the concentration of calcium and phosphate in the immediate vicinity of the implanted material may be low due to pyrophosphate, which is considered an inhibitor of hydroxyapatite formation. ALP is an enzyme that hydrolyzes this inhibitor, facilitating the production of its constituent elements [145].

In order to highlight the ability of the investigated materials to support the early stage of the osteogenic differentiation of the MC3T3-E1 preosteoblasts, the activity of the intracellular alkaline phosphatase and the level of collagen production were studied, and the obtained results are presented in Figure 11a,b, respectively.

As we can see in Figure 11a, the ALP activity exhibited by the MC3T3-E1 cells grown on the surface of all analyzed materials increased from 7 to 14 days of culture, suggesting their continuous pro-osteogenic potential. Note that after 7 days of preosteoblast incubation, the activity of this enzyme did not show significant differences between the analyzed substrates, while after 14 days of culture, a reduction in this differentiation marker intracellularly expressed by the preosteoblasts grown in contact with the HAp-based coatings was revealed. These low levels of ALP activity could suggest osteoblast maturation into osteocytes, known to express lower alkaline phosphatase levels [146].

Importantly, the coatings doped with Sr, i.e., H-Sr and H-Sr-Ag, showed low levels of ALP activity compared to cp-Ti, but these differences were not statistically significant. At the same time, in comparison with the H coatings, it could be observed that Sr supported the activity of ALP, inhibiting the adverse effects of Ag. The expression of the other early osteogenic marker, collagen, as the major component of bone extracellular proteins [147], was evinced by Sirius Red staining. Thus, after 2 and 4 weeks of preosteoblast incubation, no significant differences between cp-Ti and HAp-based coatings were found (Figure 11b). However, H-Sr and H-Sr-Ag showed significantly reduced and significantly increased collagen synthesis and deposition on compared with the cp-Ti samples. Consequently, the presence of Sr within the coatings enhanced the osteogenic capacity of MC3T3-E1 cells even in the presence of Ag. One of the important characteristics of the hard tissues of vertebrates is ECM mineralization [148], as a late marker of osteogenic differentiation [42].

ECM mineralization was determined by highlighting and quantifying calcium deposits (Figure 11c). There were no significant differences between the HAp-based coatings, except for the uncoated Ti, which showed a significantly lower value. Thus, the OD reflecting the size of the calcium deposits decreased in the following order: H ≥ H-Sr-Ag > H-Sr > H-Ag > Ti.

These results could be correlated with the results obtained following the immersion of the coated samples and cp-Ti in SBF, showing that, after 21 days, the amount increased in the same order, indicating that all types of coating supported the deposition of calcium phosphate. According to the presented results, although cp-Ti supported cell viability and proliferation, in the long term it did not support the formation of calcium deposits, a key factor in the osseointegration of implants.

In conclusion, the above in vitro studies to establish the osteogenic effect of MC3T3-E1 preosteoblasts revealed the ability of the HAp-based coatings to better support cellular differentiation towards mature osteocytes, with the beneficial effects of Sr as a single doping element or in combination with Ag being obvious.

## 4. Conclusions

The results of the present study highlighted the following:The pulse deposition technique is an optimal electrochemical technique to obtain HAp-based coatings undoped and doped with Ag and/or Sr with proper in vitro behavior.The pulse deposition technique can be successfully used to obtain Ag- and/or Sr-(co-)doped HAp coatings.Ag or/and Sr as (co-)doped elements led to a higher deposition rate and thus to thicker layers.All HAp-based coatings increased the surface energy of the substrate, and the co-doped hydroxyapatite reached the highest value.All proposed coatings enhanced the electrochemical behavior of the Ti substrate in SBF media.Even if used independently as doping elements, Ag and Sr led to a reduction in biomineralization, while when used as co-doped elements, a slight increase in the mass gain was noted when immersed in SBF, in comparison with the undoped HAp; on the other hand, when immersed in PBS, the co-doped hydroxyapatite registered the greatest decrease in mass, thus indicating more accelerated biodegradation.The cell-culture-based studies to evince the suitability of the developed HAp-based coatings for biomedical applications involving bone regeneration showed appropriate cell adhesion and proliferative potential, as well as an enhanced potential to promote the osteogenic differentiation of MC3T3 preosteoblasts; note that the presence of Sr within the HAp-based coatings, either as a unique doping element or in combination with Ag, exerted beneficial effects on the cellular response.

## Figures and Tables

**Figure 1 materials-16-05428-f001:**
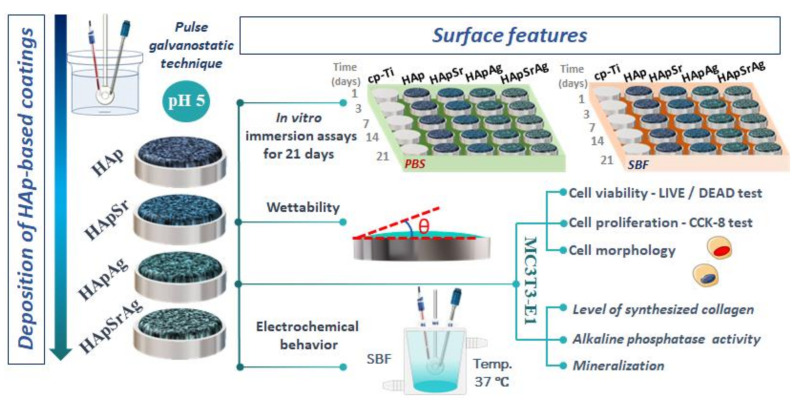
Schematic illustration of the obtained coatings and surface features.

**Figure 2 materials-16-05428-f002:**
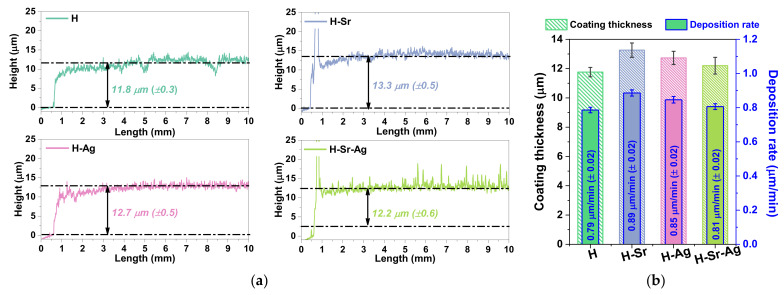
Representative profiles used to measure the coating thickness (**a**) and the deposition rate evolution for each coating (**b**).

**Figure 3 materials-16-05428-f003:**
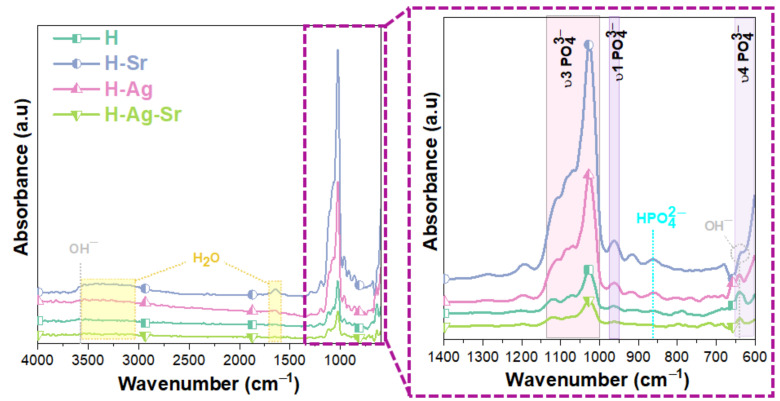
FTIR spectra of undoped and doped HAp-based coatings.

**Figure 4 materials-16-05428-f004:**
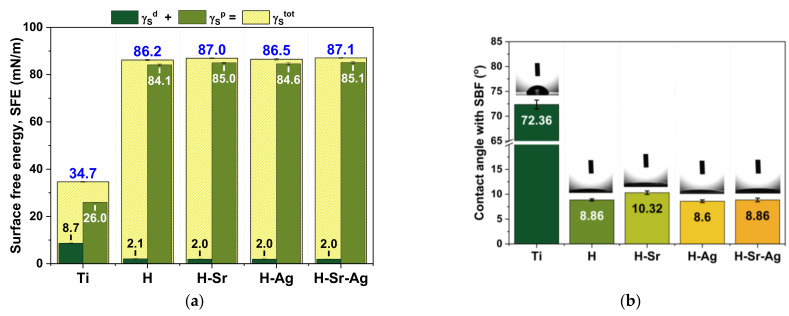
Surface free energy (**a**) and contact angle with SBF (**b**) for the investigated surfaces.

**Figure 5 materials-16-05428-f005:**
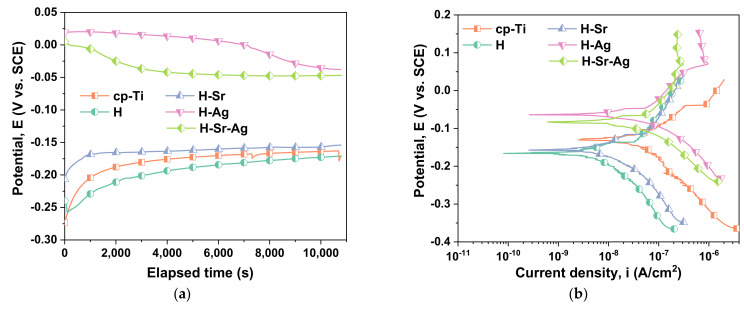
Open-circuit potential (**a**) and Tafel (**b**) plots of the investigated surfaces.

**Figure 6 materials-16-05428-f006:**
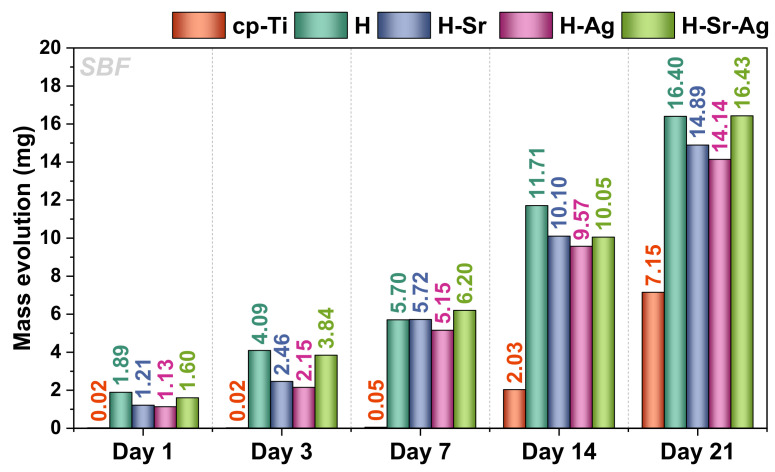
The mass evolution for cp-Ti and HAp-based coatings exposed in SBF.

**Figure 7 materials-16-05428-f007:**
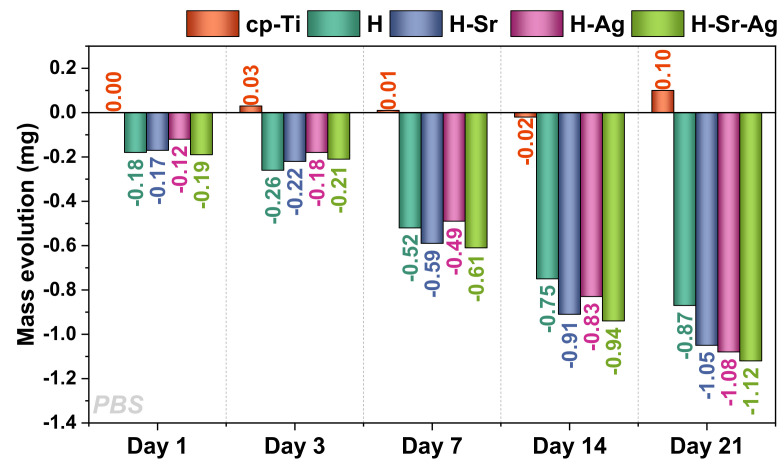
The mass evolution for cp-Ti and HAp-based coatings exposed to PBS.

**Figure 8 materials-16-05428-f008:**
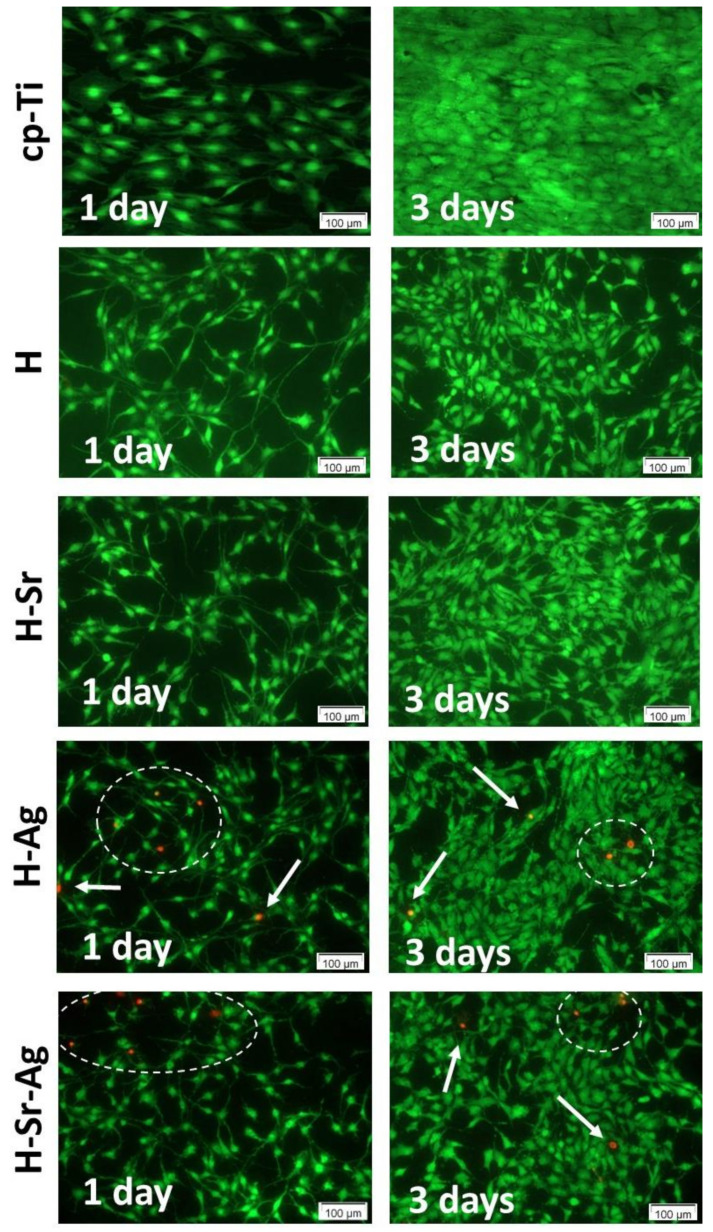
Fluorescence microscopy images of MC3T3-E1 preosteoblasts grown in direct contact with the bare cp-Ti substrate and the developed coatings for 1 and 3 days. LIVE/DEAD viability assay: live cells (green fluorescence); dead cells (red fluorescence, arrows and area delimited by the dashed line).

**Figure 9 materials-16-05428-f009:**
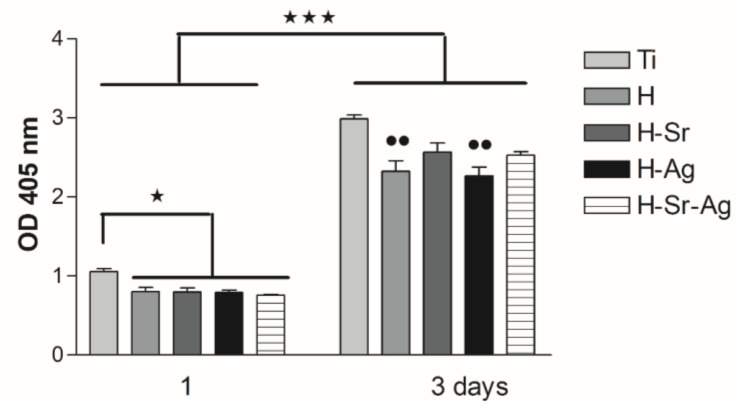
CCK-8 assay showing the proliferative capacity of cells grown for 1 and 3 days in contact with the developed HAp-based coatings and the bare cp-Ti (*** *p* < 0.001 vs. 1 day; * *p* < 0.05 for H vs. Ti coatings at 1 day; ●● *p* < 0.01 for H and H-Sr vs. Ti at 3 days).

**Figure 10 materials-16-05428-f010:**
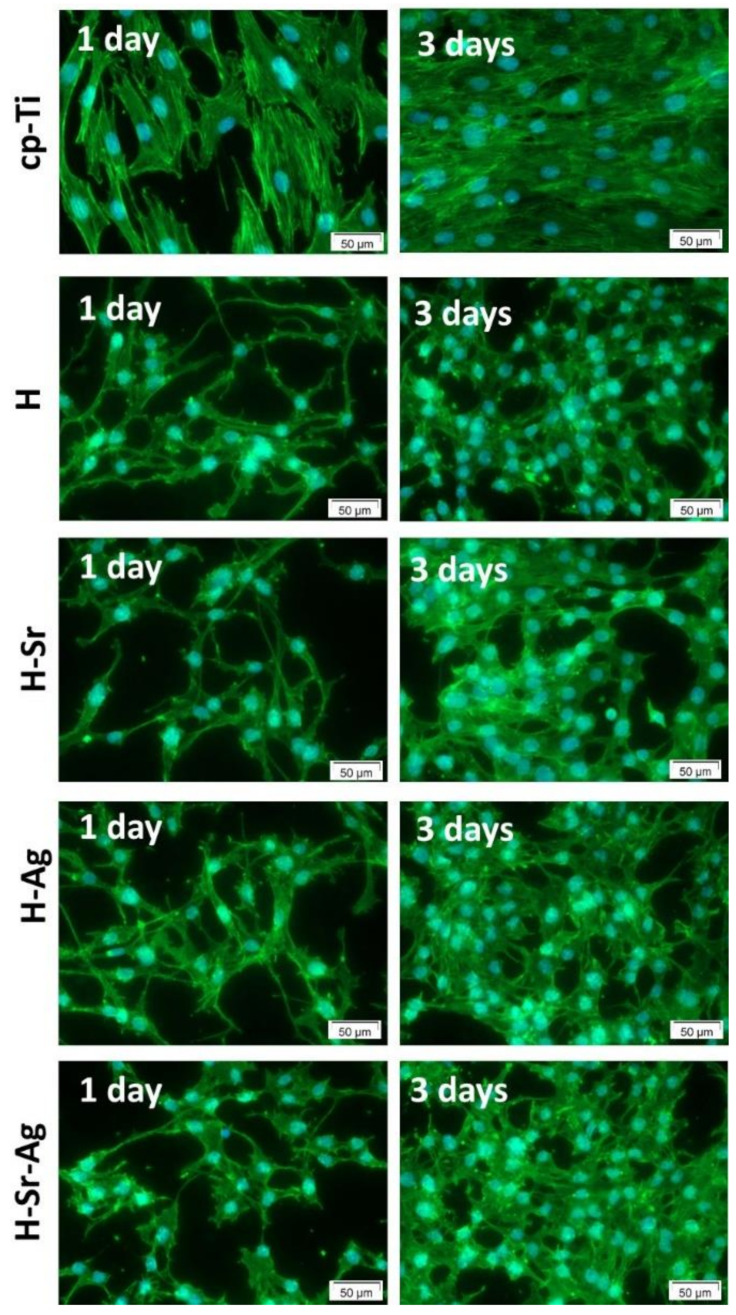
Morphology of MC3T3-E1 preosteoblasts grown in direct contact with HAp-based coatings and the control support (cp-Ti), as revealed by the Alexa Fluor 488-coupled phalloidin labeling of actin filaments (actin—green, nucleus—blue).

**Figure 11 materials-16-05428-f011:**
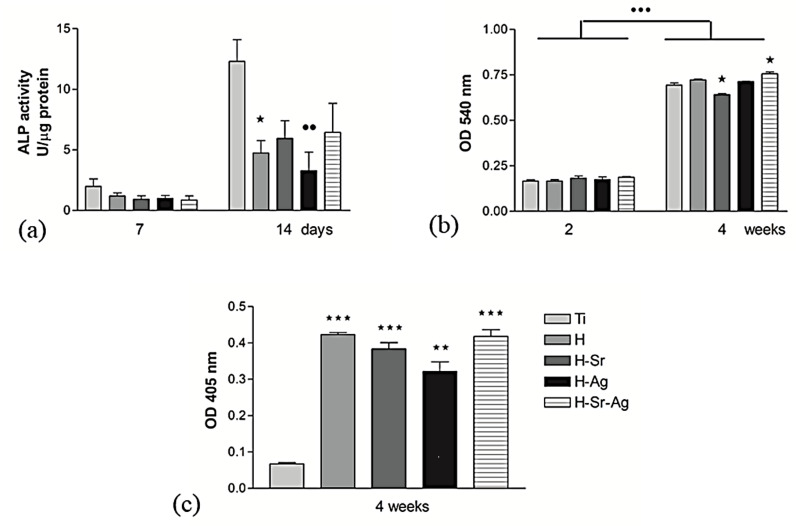
Differentiation potential of MC3T3-E1 preosteoblasts grown in contact with bare and HAp-coated cp-Ti: (**a**) specific enzyme activity of the intracellular ALP at 7 and 14 days after seeding (* *p* < 0.05 for H-Sr vs. Ti sample and ●● *p* < 0.01 for H-Ag vs. Ti at 14 days); (**b**) the level of collagen synthesis and extracellular matrix deposition after 3 weeks of culture (●●● *p* < 0.001 vs. 2 weeks; * *p* < 0.05 for H-Sr and H-Sr-Ag vs. Ti at 4 weeks); (**c**) the level of matrix mineralization as quantified by Alizarin Red staining at 4 weeks post-seeding (*** *p* < 0.001 for H-Sr and H-Sr-Ag vs. Ti, ** *p* < 0.01 for H-Ag vs. Ti).

**Table 1 materials-16-05428-t001:** Coating descriptions and codifications, chemical composition of the electrolyte, and electrochemical parameters used.

Substrate	Coating		Electrolyte (mM)				
	Description	Codification	Ca(NO_3_)_2_·4H_2_O	NH_4_H_2_PO_4_	Sr(NO_3_)_2_	AgNO_3_	pH
cp Ti	Hydroxyapatite undoped	H	10	6	0	0	5
	Hydroxyapatite doped with Sr	H-Sr	9		1	0	
	Hydroxyapatite doped with Ag	H-Ag	9.98		0	0.02	
	Hydroxyapatite co-doped with Sr and Ag	H-Sr-Ag	8.98		1	0.02	
Electrochemical parameters							
1 cycle	Activation	i_ON_	−0.85 mA/cm^2^				
		t_ON_	1 s				
	Relaxation	i_OFF_	0 mA/cm^2^				
		t_OFF_	2 s				
Number of applied cycles			900 cycles				
Deposition temperature			75 °C (±0.5 °C)				

**Table 2 materials-16-05428-t002:** Surface free energy and its components for the probe liquids in mN/m at 20 °C.

Surface Tension	Deionized Water	Ethylene Glycol	Toluene
$\gamma_{L}^{tot}$ (mN/m)	72.8	48.0	28.4
$\gamma_{L}^{d}$ (mN/m)	21.8	29.0	26.1
$\gamma_{L}^{p}$ (mN/m)	51.0	19.0	2.3

**Table 3 materials-16-05428-t003:** Composition of the acellular media used for testing the in vitro behavior.

Composition	SBF	PBS
NaCl	8.035 g/L	8 g/L
NaHCO_3_	0.355 g/L	-
KCl	0.225 g/L	0.8 g/L
K_2_HPO_4_·3H_2_O	0.231 g/L	-
MgCl_2_·6H_2_O	0.311 g/L	-
1 M HCl	3.2 mL	-
CaCl_2_	0.292 g/L	-
Na_2_SO_4_	0.072 g/L	-
Tris	6.118 g/L	-
NaH_2_PO_4_	-	1.42 g/L
pH	7.4	7.4

**Table 4 materials-16-05428-t004:** Contact angles of cp-Ti and HAp-based coatings.

Samples	DW	EG	T
	CA (°)		
Ti	67.91 *(±0.93)*	54.03 *(±0.85)*	8.08 *(±0.48)*
H	10.12 *(±0.13)*	8.55 *(±0.22)*	8.28 *(±0.79)*
H-Sr	8.67 *(±0.87)*	8.55 *(±0.74)*	8.95 *(±0.26)*
H-Ag	10.39 *(±0.35)*	10.33 *(±0.4)*	8.55 *(±0.09)*
H-Sr-Ag	8.45 *(±0.12)*	9.31 *(±0.77)*	7.45 *(±0.47)*

**Table 5 materials-16-05428-t005:** Electrochemical parameters of the investigated samples (E_corr_—corrosion potential; i_corr_—corrosion current density; Rp—polarization resistance).

Samples	E_OC_ (mV)	E_corr_ (mV)	i_corr_ (nA/cm^2^)	β_c_ (mV)	β_a_ (mV)	Rp (kΩ × cm^2^)	Pe (%)
cp-Ti	−170.38	−129.79	83.54	189.974	123.87	390.24	-
H	−171.07	−166.24	18.07	220.93	136.73	2032.17	78.4
H-Sr	−154.12	−161.07	26.27	196.54	146.89	1391.28	68.5
H-Ag	−37.88	−63.20	52.63	103.21	165.55	525.20	37.0
H-Sr-Ag	−46.93	−84.42	48.63	120.37	200.22	672.12	41.8

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
