# Peer review of "In Vitro Evaluation of Ag- and Sr-Doped Hydroxyapatite Coatings for Medical Applications"

_materials, 2023, doi:10.3390/ma16155428_

Round 1

Reviewer 1 Report

The article entitled "In vitro evaluation of Ag- and Sr-doped hydroxyapatite coatings for medical applications" has been submitted for publication in the journal Materials.

The research deals with the surface modification of pure titanium to make it bioactive for bone implant applications. The coating is produced by electrodeposition in aqueous solution. More specifically, the authors described the partial substitution of the deposited calcium phosphate coating by strontium and/or silver to improve the biological properties of the implant surface.

The obtained coatings are characterized by profilometry, FTIR spectroscopy, contact angle, and surface free energy measurements. Electrochemical characterizations (open circuit potential and potentiodynamic polarization) are carried out to study the corrosion behavior of the implant. The mass variation of the samples is studied after immersion in physiological solutions (SBF and PBS) from 1 to 21 days. The bioactivity of the coating is assessed in contact with MC3T3-E1 preosteoblasts cells (viability, proliferation, morphology, and differentiation).

The results show improved biological properties thanks to the bioactive coating and thanks to the substitution with strontium and silver.

The article is well written, the topic is well introduced, and the description of the results is good.

However, some corrections are necessary to make the article publishable in the journal Materials.

I recommend the major correction of the following points:

  - In the whole manuscript, the authors use the word "hydroxyapatite" to describe the deposited material. However, according to the deposition conditions and the experimental characterizations, the coating is probably made of calcium-deficient apatite rather than stoichiometric hydroxyapatite. This should be mentioned somewhere in the text.

  - In line 75, “Ag+” is probably more usual than “Ag2+”.

  - In Table 1, please check and confirm the values “9.98” and “0.2”. Are they good or should they be replaced by “9.8” or “0.02”?

  - In section 3.1, there is no discussion, only a description of experimental results. It is mentioned that "the doping elements influence the nucleation and crystal growth rate of HAp." How does it work? What is the mechanism? It should be discussed further.

  - In Figure 5a, OCP is measured for 3 hours. Is there any surface change during the 3 hours of immersion? Has the bioactivity process started?

  - Is there any impact of the amount of strontium and/or silver on the bioactivity of the coatings? This should be discussed.

The English language is good. Only minor corrections are necessary.

Reviewer 2 Report

In this manuscript, the authors have investigated the composite HAp-based coatings. Before I recommend it for acceptance, there are some minor improvements that could improve the quality of the manuscript.

 1.    In line 305, the authors assume that the difference in coating thickness is due to the incorporation of doping elements. Why not perform a transmission electron microscopy test for the elemental analysis or provide the crystal structure by XRD?

2.    The authors mentioned the coating thickness and the average surface roughness. If possible, AFM is able to obtain these data directly and perform relevant analyses.

Reviewer 3 Report

Dear authors, thank you for presenting interesting results on the formation of strontium-silver-containing hydroxyapatite coatings on titanium obtained by the electrochemical method.

Despite the good scientific style of the presentation of the results, as well as the generally interesting data, several important remarks should be made.

It is not clear what the authors mean by the phrase “Even though the substitution mechanism is still not fully understood”, what other features of the mechanism of substitution of calcium ions in the HA structure should be clarified.

Figure 1 - The designation of the apatite coating with the letter H is awry, it is worth using the abbreviation HAp. If such designations lead to too long designation HAP-Sr-Ag, then it is worth removing the hyphens in the indicated designations (HApSrAg).

Sodium ions were present in the solution (sodium hydroxide was used to maintain pH = 5). Is sodium present in the coatings? It is known that the structure of apatite allows sodium ions to enter the structure quite easily.

No XRD data presented, why do the authors think that a HAp coating is formed at pH 5? In addition, the FTIR spectra shown definitely lack the peak of the hydroxyl group at 3570 cm-1, which is characteristic of the formed HAp structure.

In addition, regarding the results of IR and the possible inclusion of sodium ions in the structure of the coating, the authors note the presence of peaks assigned to carbonate groups. It is known that when sodium ions enter the HA structure, at the same time, part of the phosphate groups can be replaced by carbonate groups to compensate for the charge balance. The authors do not consider the possible type of carbonate incorporation into the structure (A, B, AB type, or the carbonate peaks can be associated with air CO2 adsorbed on the surface), which may clarify a more detailed analysis of the FTIR spectra.

There are also no results of elemental analysis of the coating composition: the Ca/P ratio in solution equal to 1.67 will not necessarily remain in the coating, especially at pH = 5. It is also interesting in what quantities other ions are present in the coating (strontium, silver, sodium).

Of their remarks on the design, it should be noted that the authors do not always indicate the doi of the references, although in most of the missed cases they exist.

Reviewer 4 Report

The authors prepared HAp coatings undoped and doped with Ag and/or Sr on titanium. Various properties of the coatings were characterized in detail. The work is interesting. However, the current manuscript cannot be published. Some  revisions are needed.

1. The unit of Ca(NO3)2, NH4H2PO4, etc. listed in Table 1 should be provided.

2. The authors stated that “the amount of mass lost increases with the increase in porosity of coatings” in Page 14 Line 521. The authors should provide the porosity values of different coatings.

3. The authors are encouraged to provide the SEM images of the different coatings.

4. The current data provided in the manuscript is not sufficient to confirm that the coatings are HAp. The authors should provide the XRD patterns of different coatings.

5. The conclusion can be further refined.

The writing of the manuscript is well written.

Round 2

Reviewer 1 Report

The authors have appropriately modified their manuscript according to my comments.

I recommend acceptance.